# A Novel Isaindigotone Derivative Displays Better Anti-Proliferation Activities and Induces Apoptosis in Gastric Cancer Cells

**DOI:** 10.3390/ijms23148028

**Published:** 2022-07-21

**Authors:** Kangjia Du, Chengjie Yang, Zhongkun Zhou, Yunhao Ma, Yanan Tian, Rentao Zhang, Hao Zhang, Xinrong Jiang, Hongmei Zhu, Huanxiang Liu, Peng Chen, Yingqian Liu

**Affiliations:** School of Pharmacy, Lanzhou University, 199 Donggang West Road, Lanzhou 730020, China; dukj19@lzu.edu.cn (K.D.); yangchj18@lzu.edu.cn (C.Y.); zhouzhk18@lzu.edu.cn (Z.Z.); mayh2019@lzu.edu.cn (Y.M.); tianyn19@lzu.edu.cn (Y.T.); zhangrt19@lzu.edu.cn (R.Z.); zhanghao2020@lzu.edu.cn (H.Z.); jiangxr20@lzu.edu.cn (X.J.); hmzhu@lzu.edu.cn (H.Z.); hxliu@lzu.edu.cn (H.L.)

**Keywords:** gastric cancer, isaindigotone, anti-proliferation, mitochondrial potential

## Abstract

Isaindigotone is an alkaloid containing a pyrrolo-[*2,1-b*]quinazoline moiety conjugated with a benzylidene group and isolated from the root of *Isatis indigotca* Fort. However, further anticancer activities of this alkaloid and its derivatives have not been fully explored. In this work, a novel isaindigotone derivative was synthesized and three different gastric cell lines and one human epithelial gastric cell line were used to study the anti-proliferation effects of the novel isaindigotone derivative BLG26. HGC27 cells and AGS cells were used to further explore the potential mechanisms. BLG26 exhibited better anti-proliferation activities in AGS cells with a half-maximal inhibitory concentration (IC_50_) of 1.45 μM. BLG26 caused mitochondrial membrane potential loss and induced apoptosis in both HGC27 cells and AGS cells by suppressing mitochondrial apoptotic pathway and PI3K/AKT/mTOR axis. Acute toxicity experiment showed that LD_50_ (median lethal dose) of BLG26 was above 1000.0 mg/kg. This research suggested that BLG26 can be a potential candidate for the treatment of gastric cancer.

## 1. Introduction

According to the latest cancer statistics, gastric cancer (GC) is the sixth most common cancer and accounts for 5.6% of overall cancer cases, with an estimated mortality of 769,000 cases in the world [1]. As the most effective treatment for patients with advanced metastases, chemotherapy often has no therapeutic effect during the treatment process due to intrinsic or acquired resistance, which directly induces treatment failure [2]. Thus, novel chemotherapeutic agents are urgently needed for more efficient treatment for gastric cancer patients.

*Isatis indigotica* is a cruciferous herb and is widely distributed in northern and central China [3]. Previous study indicated that methanolic extracts of the roots of *Isatis indigotica* could reduce inflammatory mediators’ production [4]. The aqueous extract of the roots of *Isatis indigotica* was also reported to protect C57BL/6J mice from lipopolysaccharide (LPS)-induced sepsis [5]. N-butanol extract from Folium isatidis was reported to inhibit the expression of pro-inflammatory factors [6]. Clinical studies have found that gastric cancer tissues are accompanied by different degrees of inflammation, cell infiltration, and increased expression of inflammatory factors [7,8,9,10]. Thus, anti-inflammation drugs could be preventative agents in various cancers induced by carcinogen. Agents with anti-inflammation activities may also have chemotherapeutic potential in cancer treatment.

Isaindigotone was isolated and purified from *Isatis indigotica*, which was reported to possess many pharmacological activities, such as antibiotic, anti-endotoxin, anti-inflammatory, and anti-tumor activities [11,12]. Previous studies showed that isaindigotone could suppress inflammatory reactions in lipopolysaccharide-induced BV-2 cells [13]. Huang’s research also revealed the anti-proliferation activities of isaindigotone derivatives [14,15]. Studies have shown that the PI3K/AKT/mTOR pathway is crucial in a variety of inflammatory diseases such as chronic obstructive pulmonary disease (COPD), inflammatory bowel disease (IBD), multiple sclerosis (MS), and systemic lupus erythematosus (SLE), etc. [16,17,18]. One recent study showed that the PI3K–AKT signal pathway was included in the medicinal efficacy of *Isatis indigotic* [19]. This research demonstrated the potential anti-inflammatory and anti-proliferation activities of isaindigotone and its derivatives, but whether isaindigotone and its derivatives can affect the PI3K/AKT/mTOR signal pathway is unclear.

In this work, a novel isaindigotone derivative, BLG26, was synthesized. To explore the cytotoxic activities of BLG26 on the proliferation of four different gastrointestinal cancer cell lines (gastric cancer cell AGS, hepatoma carcinoma cell SMMC-7721, pancreatic cancer cell PANC-1, colorectal cancer cell HCT116), MTT assay was performed. Due to significant cytotoxity activities in gastric cancer cell line, three gastric cancer cell lines (AGS, SGC7901, HGC27) were then used to determine the cytotoxic activities. The pharmacological mechanism of BLG26 was further explored in vitro and its acute toxicity was also evaluated. Our work may provide a potential anti-gastric cancer agent in future.

## 2. Results

### 2.1. Identification, Characterization, and Chemical Synthesis of BLG26

^1^H NMR, ^13^C NMR, HRESIMS, and HPLC Spectra of BLG26 are shown in Appendix A.

### 2.2. Anti-Proliferation Activity of BLG26

In vitro cytotoxic activities of BLG26 were investigated on four human gastrointestinal cancer cell lines (AGS, HCT116, SMMC-7721, and PANC-1) with MTT assay. The IC_50_ values of BLG26 and positive control, cisplatin (CIS, CIS was used as a first-line treatment for gastric cancer), are shown in Table 1. BLG26 showed excellent cytotoxicity in gastric cancer cell line AGS (IC_50_ = 1.45 μM).

Due to the excellent cytotoxicity of BLG26 on gastric cancer cell line, three gastric cancer cell lines were then used to further investigate its anti-proliferation activities. As was shown in Figure 1, the cell viabilities of the three gastric cancer cell lines were significantly inhibited in dose- and time-dependent manner. The inhibitory effects of BLG26 were also assayed on gastric epidermal cell line (GES-1). As observed, BLG26 showed good safety on normal gastric cell GES-1 (Figure 1D). In conclusion, BLG26 had better cytotoxicity in gastric cancer cell lines, especially in AGS (IC_50_ = 1.45 μM, 48 h) and HGC27 cell lines (IC_50_ = 1.67 μM, 48 h). Thus, AGS and HGC27 cell lines were chosen for further mechanism study.

### 2.3. BLG26-Induced Slight Cell Cycle Arrest

The effects of BLG26 on cell cycle were also detected by flow cytometry. Both AGS cells and HGC27 cells were treated with BLG26 for 24 h. The result showed that (Figure 2) BLG26 induced slight G2/M arrest in AGS cells. The population of G2/M phase increased 8.2% when compared with control group. However, after being treated with BLG26, in comparison with control group, the proportions of different phases in HGC27 cells had little variations. Therefore, the anti-proliferation effects of BLG26 on AGS cells and HGC27 cells can not be caused by cell cycle arrest.

### 2.4. Morphological Analysis by Hoechst 33258 Staining

To determine the mechanism of how BLG26 affected AGS and HGC27 cells, Hoechst 33258 was used to stain the treated cells to analyze whether its effects are related to apoptosis. Cell nuclei containing condensed or fragmented chromatin were considered as apoptosis cells. Hoechst 33258 staining images indicated that BLG26 could cause apoptosis in both AGS cells and HGC27 cells in a dose-dependent manner (Figure 3).

### 2.5. BLG26-Induced Apoptosis in AGS and HGC27 Cells

The morphological changes of the cells caused by BLG26 remind us to employ Annexin-V/PI staining. The result exhibited significant apoptosis rates that were induced by BLG26 (Figure 2). Especially, the apoptotic cells were increased to 50.4% and 22.6% when treated with 3 μM BLG26 on AGS cells and HGC27 cells, respectively. As the variation of the apoptotic cells were changed in a dose-dependent manner, consistent with the condensed and fragmented nucleus observed in Hoechst 33258 staining, it can be concluded that BLG26 can induce apoptosis in AGS cells and HGC27 cells.

### 2.6. BLG26-Triggered Mitochondrial Membrane Potential (MMP) Loss in AGS Cells

To further investigate the mechanism causing apoptosis activities of BLG26, MMP changes were measured using JC-10 as the probe. CCCP was used as positive control and DMSO was used as negative control. Generally, red fluorescence represents an aggregate state of JC-10 in normal cells while green fluorescence represents a monomeric state of JC-10 in apoptosis cells. High density of red fluorescence could be observed in DMSO-treated AGS and HGC27 cell. As the concentration of BLG26 increased, more and more AGS and HGC27 cells emitted greater green fluorescence, which indicated that loss of MMP occurred among the majority of AGS cells and HGC27 cells (Figure 4). Thus, BLG26 can also cause MMP loss.

### 2.7. Molecular Docking Study

The crystal structure of AKT1 (PDB:6HHF) was downloaded from the RCSB PDB database and the protein was minimized and initially optimized in the Protein Prep module of Schrödinger software for the complementation of missing residue side chains, protonation, dehydration, and hydrogenation under the OPLS2005 force field. Lattice files were then generated by identifying ligands in the crystal structure. For small molecule ligands, we optimized the small molecule dataset in the OPLS_2005 force field using the LigPrep module in Schrödinger software, and finally used the Glide module for docking. According to previous studies, isaindigotone derivatives have better affinity with AKT1. In the docking analysis, tyrosine 18, 80, 263, 210, 213, leucine 264, alanine 212, valine 270, and isoleucine 84 of protein AKT1 were bound to BLG26 by hydrophobic interactions. As polar amino, glutamine 79, threonine 82, 211, tryptophan 205, and histidine 207 formed polar interactions with BLG26. Arginine 273, 206 and lysine 268, as basic amino acids (positively charged in a solvent), form a positive interaction with BLG26 due to its basicity. Aspartic acid 292 (negatively charged in the solvent) has a negative interaction due to its acidity. It is worth noting that according to the 2D interaction diagram of 6HHF and BLG26 (Figure 5), tyrosine 272 forms a hydrogen bond with the amino group in the BLG26, and aspartic acid 274 also forms a salt bridge with the amino group at this position. This was also confirmed in the interaction diagram of small ligand molecules and pocket residues, which greatly stabilizes the binding of small ligand molecules to the receptor.

### 2.8. Apoptosis-Related Proteins Regulated by BLG26

As shown in Figure 6, we then investigated the effects of BLG26 on PI3K/AKT/mTOR pathway. The phosphorylation forms of AKT and mTOR kinase and expression of PI3K was reduced when treated with BLG26 in both AGS and HGC27 cells. Western blot analyses also revealed that the expression of pro-apoptotic proteins Bax and cleaved-caspase 3 was up-regulated and the expression of the anti-apoptotic protein Bcl-2 was down-regulated correspondingly in both cell lines after treatment with BLG26. Therefore, BLG26 could not only participate in the regulation of the PI3K/AKT/mTOR signal pathway, but it also affected the expression of mitochondria-related apoptotic proteins.

### 2.9. The Preliminary Acute Toxicity

In the acute toxicity trial, 30 Kunming mice (female) were divided into five groups and administrated by oral gavage at 5.0, 50.0, 500.0, and 1000.0 mg/kg (single dose and control groups). Curling up and hypokinesia only happened in high administrative group and the mouse recovered in five to seven days. In 1000.0 mg/kg group, two mice died on the fifth day after administration. Besides that, there was no more abnormal behavior and death among the five groups during the 14 days of observation, which indicated that the maximum tolerated dose for the mouse was more than 500.0 mg/kg and less than 1000.0 mg/kg, and the LD50 was more than 1000.0 mg/kg. The pathological analysis of their organs showed that there was no obvious abnormality in all groups except moderate edema of liver cells (1000.0 mg/kg). Therefore, BLG26 has good oral tolerance in mouse (Figure 7).

### 2.10. BLG26 Inhibited Cell Migration

The effects of BLG26 on cell migration were then examined by transwell. As shown in Figure 8, BLG26 significantly reduced cell migration in dose-dependent manner in both AGS and HGC27 cells.

## 3. Discussion

As a relevant disease, gastric cancer caused great health burden in China [20]. Natural products and their derivatives are believed to be crucial sources in anticancer drugs development [21]. In this study, we demonstrated that BLG26, a novel synthesized isaindigotone derivate, exhibited potential anti-proliferation effects on gastric cancer cells via causing apoptosis through PI3K/AKT/mTOR and mitochondrial pathway. The findings showed that BLG26 is a potential therapeutic candidate for gastric cancer treatment.

In the present study, the anti-proliferation activities of BLG26 were evaluated on AGS, HCT116, SMMC-7721, and PANC-1 cell lines. BLG26 showed the best cytotoxic activity on the gastric cancer cell line, AGS, with IC_50_ values of 1.45 μM. MTT assay also showed that BLG26 has good anti-proliferation activities in other gastric cancer cell lines (SGC7901, HGC27). When compared with Pt, BLG26 showed less cytotoxic effect on human epithelial cells (GES-1).

In the process of apoptosis, morphological changes including cell shrinkage, chromatin condensation, and plasma membrane blabbing can be observed [22]. Here, after being treated with BLG26, the results of Hoechst stain assay showed that the cells exhibited typical morphological changes. In addition, the results of flow cytometry analysis further confirmed that BLG26 induced the apoptosis in AGS and HGC27 cells. Moreover, besides apoptosis, cell cycle arrest at different phases can also affect cell proliferation [23]. Thus, cell cycle analysis was also performed with flow cytometry and the results indicated that BLG26 could only cause slight cell cycle arrest in AGS cell line.

Previous studies proved that the dysfunction of mitochondria plays an important role in promoting apoptosis [24]. Mitochondrial potential loss causes translocation of Cyt C, which further triggers the activation of caspase-9 and caspase-3 and induces apoptosis [25]. The results observed with fluorescence microscopy showed that BLG26 decreased the membrane potential in a dose-dependent manner. The increase in activated caspase-3 was also consistent with this. Furthermore, mitochondrial membrane permeabilization was also reported to be affected by Bcl-2 family members [26]. The central pro-apoptotic member, Bax, can induce mitochondrial dysfunction, while Bcl-2 can block the dysfunction [27]. Therefore, the variation of Bax and Bcl-2 can also prove whether BLG26 can induce cell apoptosis. BLG26 markedly increased Bax and decreased Bcl-2, which further proved that BLG26 promoted cell apoptosis through mitochondrial apoptotic pathway.

PI3K/AKT/mTOR signaling pathway (Figure 8C) plays a vital role in many aspects of cell growth, metastasis, and cell death regulation [28]. Thus, this pathway has become a hot spot in drug development. As a serine- and threonine-specific protein, AKT is also known as PKB (protein kinase B), which can be commonly found in all human cell types [29]. A recent study revealed that AKT kinase is a crucial target in small molecular drug discovery for its role in hyperactivation, cell proliferation, and cell survival in many human cancer types [30]. AKT oncogene was preliminary identified, fully cloned, and characterized in the last century (1987–1991) [31]. The first AKT inhibitor, ML-9, could induce autophagy by stimulating the formation of autophagosomes and inhibiting their degradation (2000) [32]. NL-71101 was designed ased on the structure of kinase A (PKA) inhibitor H-89 and was found to have better activity (2002) [32]. The results of a phase-I clinical trial of the AKT inhibitor perifosine was reported in 2004 [33]. One recent study showed that AKT inhibitor could enhance taxol-caused ovarian cancer cell inhibition [34]. In general, many ATP-competitive inhibitors, such as ipatasertib and capivasertib, and allosteric AKT inhibitors like miransertib and MK-2206, have been developed. The maintaining of AKT kinase’s inactive conformation is mainly dependent on its PH domain [35]. Many small molecule agents were designed based on target binding and could interact with different residues in PH domain [36]. Besides that, AKT1 and AKT2 nanobodies were developed to target AKT1 and AKT2 separately [37]. In addition, PROTAC (proteolysis targeting chimera) was also developed to degrade AKT kinase [38]. For now, many clinical trials of AKT inhibitors are underway [39]. AKT inhibitors like capivasertib and ipatasertib are being tested in a phase-Ⅲ clinical trial. Research also showed that combination of paclitaxel and ipatasertib could increase the overall survival of TNBC (triple negative breast cancer) patients. Previous studies indicated that PI3K/AKT/mTOR signal pathway alteration was found in 47% gastric cancer cases [40]. It was reported that downregulation of AKT induces apoptosis in lung cancer [41]. Studies also indicated that inhibition of AKT affected cancer cell migration [42]. The results of transwell assay showed that BLG26 inhibited cell migration, and BLG26 can cause significant decrease of PI3K, phosphorylated AKT, and phosphorylated mTOR. Overall, BLG26 can affect mitochondrial membrane potential and PI3K/AKT/mTOR pathway. In vivo toxicity was also determined using female Balb/c mice. The results showed that oral toxicity of BLG26 is very low. All this evidence suggests that BLG26 exerts good anti-proliferation activity and minor toxicity in vivo.

The study of the parent structure of BLG26 (isaindigotone) and its derivatives can also provide references for its further applications. The adaptive structural feature of isaindigotone enabled the aromatic ring to bind with G-quadruplex. Tan’s team synthesized and characterized four isaindigotone derivatives to explore the novel selective G-quadruplex binding ligands for tumor chemotherapy. The results showed that isaindigotone derivatives are a promising G-tetrameric binding ligands with strong recognition of double-stranded DNA [43]. The high G-quadruplex selectivity and telomerase inhibition of these ligands revealed that the chemical biology and pharmacological mechanisms of isaindigotone derivatives deserve more in-depth investigation. Huang designed and synthesized a series of isaindigotone derivatives that were able to effectively block NM23-H3 binding to the G-quadruplex and significantly reduce c-myc transcription, causing cell cycle arrest and apoptosis. Their team found that the novel derivatives had better binding affinity with NM23-H3 and G-quadruplex than previously reported G-quadruplex stabilizers [14]. Chen’s team first identified the inhibitory effect of isaindigotone derivatives on Bloom syndrome protein (BLM). BLM is a DNA decapping enzyme that plays an important role in homologous recombination repair (HRR)-related regulation. His team has identified isaindigotone derivatives as novel BLM inhibitors through synthesis, screening, and evaluation. Among them, compound **29** is a potent BLM inhibitor with high binding affinity and good inhibition of BLM. Cellular experiments showed that compound **29** effectively interfered with BLM recruitment at DNA double-strand break sites, and promoted the accumulation of RAD51, and finally regulated the HRR process. Also, compound **29** significantly induced DNA damage response, apoptosis, and proliferation arrest in cancer cells [44]. PAN designed and synthesized a series of isaindigotone derivatives capable of inhibiting acetylcholinesterase [45]. Yan continued the above work by synthesizing a series of chlorine-substituted isaindigotone derivatives as acetylcholinesterase and β amyloid polymerization inhibitors. Circular dichroism and scanning electron microscopy observed that compound **6c** was able to inhibit β-fold aggregation and fibril formation [11]. Thus, further applications of BLG26 should be explored in the future.

In this study, we obtained a novel isaindigotone derivative by chemical synthesis and explored the potential anti-proliferation pathways of isaindigotone and its derivatives based on their in vitro anti-tumor cell proliferation activity. The specific binding target will be the main focus in subsequent studies. We hope to provide a more active and less toxic compound for clinical gastric cancer treatment and provide new ideas for the development of isaindigotone derivatives in the future.

## 4. Materials and Methods

### 4.1. Chemistry

^1^H NMR (400 MHz) and ^13^C NMR (100 MHz) spectra were recorded with a Bruker AM-400 (Bruker Company, Billerica, MA, USA) spectrometer at 400 and 101 MHz, respectively, and TMS was used as internal standard. Electrospray ionization mass spectrometry was recorded on ZAB-HS and Bruker Daltonics APEXII49e instruments. Melting point (mp) was determined using a Kofler melting point apparatus. The purity of BLG28 was confirmed by analytical HPLC (VARIAN, ProStar) with an Ultimate XB-C18 column (methanol:water 80:20, 0.1% TFA, 1 mL/min). All chemical reagents used in the synthesis process were purchased from Energy Chemical. All of the solvents were purified according to standard methods.

#### 4.1.1. Synthetic Procedure for Target Compound **2**

A solution of 2-amino-4, 5-difluorobezoic acid (3.34 g, 19.3 mmol) and pyrrolidin-2-one (3.0 mL, 39.5 mmol) was carefully added to 45 mL of POCl_3_ at room temperature. The mixture was then stirred at 110 °C for 7 h. After POCl_3_ was removed under reduced pressure, the residue was poured into ice water, and then a solution of NaOH was added to make the solution basic. The mixture was extracted with 50 mL portions of CH_2_Cl_2_ (3 times). The combined organic phase was dried over MgSO_4_ and concentrated under diminished pressure. The crude product was purified by using flash column chromatography with EtOAc/petroleum ether (1:4) elution to afford a white solid. The yield was 81% and the mass of compound **2** was 222.0605 (Figure 1).

#### 4.1.2. Synthetic Procedure for Target Compound **3**

An AcOH (70 mL) solution of compound **2** (10 mmol) and 4-(fluoro) benzaldehyde (20 mmol) was added to a catalytic amount of NaOAc. The mixture was then stirred at 115 °C for 6 h. After AcOH was partly removed under reduced pressure, the residue was poured into 15 mL of ice-cold acetone and then filtered and washed with acetone to afford a yellow solid.

#### 4.1.3. Synthetic Procedure for Target Compound BLG26

A DMF (18 mL) solution of compound **3** (5 mmol) and *N*-methylpiperazine (2.5 mL) was added to a catalytic amount of Na_2_CO_3_. The mixture was then stirred at 140 °C for 2 h. After DMF was partly removed under reduced pressure, the residue was poured into 30 mL of water, and then filtered and washed with water to afford a white solid.

### 4.2. Cell Lines and Cell Culture

HCT116, AGS, PANC-1 cell lines were purchased from the America Type Culture Collection. (ATCC Lot: 70019042, 70012225, 70018880). HCT116, AGS, SMMC-7721, HGC27, SGC7901 cells were cultured in Roswell Park Memorial Institute-1640 (RPMI-1640, Solarbio Invitrogen Co., Beijing, China). PANC-1 and GES-1 cells were maintained in high glucose Dulbecco’s modified Eagle medium (DMEM, Solarbio Invitrogen Corp., Beijing, China). All cells involved in this work were supplemented with 10% fetal bovine serum (FBS) and incubated at 37 °C with 5% CO_2_.

### 4.3. In Vitro Cytotoxicity Evaluation

Cells were seeded in 96-well plates (1 × 10^4^ cells/100 μL) and incubated with BLG26 at 0.25, 0.50, 1.00, 2.00, 3.00, 4.00, and 5.00 μmol/L for 48 h. DMSO was used as negative control. Then the cells were added with 10.0 μL 3-(4,5-dimethylthiazol-2-yl)-2,5-diphenyltetrazolium bromide and incubated at 37 °C for 4 h. Finally, after removing the medium, 100 μL of DMSO were added to dissolve the blue crystal. The absorbance was measured by a microplate reader (SpectraMax190, USA) at 490 nm. The IC_50_ was calculated via SPSS software.

### 4.4. Cell Cycle Analysis

Cells in log-phase (7 × 10^5^/well) were seeded on 6-well plates and treated with DMSO and different concentrations of BLG26. The cells were then fixed with 70% ethanol at 4 °C overnight. After that, the cells were stained with 40 μg/mL propidium iodide (PI) and detected by flow cytometer (BD LSRFortessa, USA). The date was analyzed by Flow Jo software (Tree Star Inc., Ashland, OR, USA).

### 4.5. Hoechst 33258 Staining

Cells were seeded on cover glasses and a series concentration of BLG26 were added into the cells and incubated for 48 h. The cells were then fixed with 4% paraformaldehyde for 30 min at room temperature and incubated with the 100:1 diluted Hoechst 33258 (Solarbio life sciences, China, 10.0 μg/mL) for 10 min. The stained cells were ultimately examined with a fluorescence microscope (Zeiss, Germany).

### 4.6. Apoptosis Analysis by Flow Cytometry

Cells were seeded on 6-well plates and incubated with BLG26 (1, 2 and 3 μM) for 48 h. Then, the cells were resuspended and diluted to 1 × 10^6^ cell/mL with binding buffer mixture (Solarbio life sciences, China, 10.0 μg/mL). After that, 5 μL Annexin-V-fluorescein was added and the mixture was incubated in the dark for 5 min at 25 °C. In the end, rapid measurements were conducted by a flow cytometer (BD LSRFortessa, USA) following adding 10 μL Propidium iodide and 400 μL PBS into the mixture.

### 4.7. Measurement of the Mitochondrial Membrane Potential (MMP) with JC-1

MMP was determined using the JC-1 kit (Solarbio Invitrogen Co., Beijing, China). After being embedded on cover glasses in 24-well plates, the cells were treated with different concentrations of BLG26 (0, 1, 2, and 3 μM) for 24 h and fixed with 4% paraformaldehyde for 40 min. The cells were then stained with JC-1 dye at 37 °C for 30 min. Carbonyl cyanide-3-chlorophenylhydrazone (CCCP, 10.0 μM) was used as positive control. Finally, the stained cells were observed with a laser confocal fluorescence microscopy (Zeiss, Germany).

### 4.8. Docking Studies

The three-dimensional structure of the AKT (PDB code: 6HHF) obtained from RCSB PDB database was used for molecular modeling. The Schrodinger 10.2 was employed for docking calculations. Pymol 2.4 was used for graphic display.

### 4.9. Western Blot Analysis

Control and BLG26-treated cells (48 h) were collected and the total cell proteins were extracted using PIPA buffer containing protease inhibitor. The solution was then centrifuged at 20,000 rpm for 15 min at 4 °C. Upper solution was obtained to analyze the changes of the protein in the sample. Then, 15% sodium dodecyl sulfate polyacrylamide gel electrophoresis (SDS-PAGE) was used to separate the proteins. The target proteins were finally transferred onto PVDF membrane and detected with a chemiluminescence analysis system.

### 4.10. The Preliminary Acute Toxicity Study of BLG26 Using Kunming Mice

The acute toxicity research complied with the National Product Administration of China issued Technical Guidelines for Acute Toxicity Experiment for Chemical Drugs. Adult female Kunming mice of 4 to 6 weeks were obtained from Animal Experiment Center of Lanzhou University [SCXK(Gan)2018-0002]. All experimental protocols were approved by Ethic Committee of Lanzhou University.

30 Kunming mice (female) were divided into five groups and administrated by oral gavage at 5.0, 50.0, 500.0, 1000.0 mg/kg (single dose, and control groups). All of them were given a single dosage of BLG26 and solvent (PEG400) by oral after a 12 h fast with free drink. Food was supplied after 2 h of treatment. Their toxic reaction and mortality were then observed. The body weight, death time, symptom, occurring time of each mouse were recorded for 14 days.

### 4.11. Transwell Cell Migration Assay

The invasion activity of the treated cells was analyzed with a transwell apparatus. Cells (3 × 10^4^) were suspended in serum-free RPMI-1640 medium and seeded into the upper chambers (8-μm pore size, Corning). Then, 600 μL RPMI-1640 medium with 20% FBS medium was added to the lower chamber and incubated for 48 h. The embedded cells were subsequently fixed with 4% paraformaldehyde for 40 min in room temperature and stained with 0.1% crystal violet for 20 min. Cells that migrated to the lower chamber were then photographed and counted under an optical microscope.

## 5. Conclusions

The present work revealed the anti-proliferation effects of BLG26 in vitro and evaluated its acute toxicity in vivo. MMP loss could be observed in treated cells. Mechanism studies revealed that BLG26 induced cell apoptosis by affecting PI3K/AKT/mTOR and mitochondrial apoptotic pathways. BLG26 could also inhibit AGS and HGC27 cell migration. Given these results, the anti-proliferation activities of isaindigotone derivatives were found to be improved through structural modification. Hence, BLG26 is a potential anti-gastric cancer agent and will be further investigated in future.

## Data Availability

The data presented in this study are available on request from the corresponding author.

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
