# Peer review of "A Novel Isaindigotone Derivative Displays Better Anti-Proliferation Activities and Induces Apoptosis in Gastric Cancer Cells"

_ijms, 2022, doi:10.3390/ijms23148028_

Round 1

Reviewer 1 Report

In this work, a novel 16 isaindigotone derivative was synthesized and three different gastric cell lines and one human epithelial gastric cell line were used to study the anti-proliferation effects of the novel isaindigotone derivative BLG26.

The paper is of interest.

First of all, I would suggest a mother-tongue revision of english language.

Secondly: it is not clear why you chose only GC cell lines? Is there experience with other cell lines?

The figure is poor, I would improve it and consider to insert also the PI3K-AKT-mTOR pathway.

Discussion is too short. Please improve it and discuss possibile clinical development of your findings? Do you see room for a phase I study or something else?

Author Response

Dear reviewer #1:

Comment 1: I would suggest a mother-tongue revision of english language.

Response 1: We truly appreciate the reviewer’s comments. We have tried our best to revise the English of the whole manuscript carefully. In addition, we also have asked colleagues who are skilled authors of English language papers to help us improve the language. We hope that the language is now acceptable for the next review process.

Comment 2: it is not clear why you chose only GC cell lines? Is there experience with other cell lines?

Response 2: Thank you for your instructive advice. We aimed to explore the cytotoxic activities of BLG26 on gastric cancer cell AGS, hepatoma carcinoma cell SMMC-7721, pancreatic cancer cell PANC-1, colorectal cancer cell HCT116. The results showed that BLG26 had better anti-proliferation activities on gastric cancer cell AGS. Thus, we chose three gastric cancer cell lines (AGS, SGC7901, HGC27) for further research.

Comment 3: The figure is poor, I would improve it and consider to insert also the PI3K-AKT-mTOR pathway.

Response 3: We appreciate the reviewer’s comments. We have re-organized some of the figures and inserted a figure to explain PI3K-AKT-mTOR pathway.

Comment 4: Discussion is too short. Please improve it and discuss possibile clinical development of your findings? Do you see room for a phase I study or something else?

Response 4: Special thanks to you for your kind suggestion. We have added more discussion about possible clinical development of isaindigotone derivative BLG26 and current therapeutic agents targeting AKT in the revised manuscript. As for the room for a I study, we need to do more research on in vivo pharmacodynamics studies to evaluate the possibilities.

Reviewer 2 Report

The article by Kangjia "Du and colleagues A Novel Isaindigotone Derivative Displays Better Anti-Prolif-2 eration Activities and Induces Apoptosis in Gastric Cancer 3 Cells" has been done with care and the figures are explicit. If it is not a choice of the scientific journal in the succession of topics I would put Materials and Methods first and then results so the text is more fluent.

Author Response

Dear reviewer #2:

Comment 1: If it is not a choice of the scientific journal in the succession of topics I would put Materials and Methods first and then results so the text is more fluent.

Response 1: We acknowledge the reviewer’s comments and suggestions very much, which are valuable in improving the quality of our manuscript. We have put Materials and Methods first and then results in the revised manuscript.

Reviewer 3 Report

Authors synthesized a novel isaindigotone derivative, BLG26, which inhibited the proliferation of four different gastrointestinal cancer cell lines. The study is easy to follow. 

Minor comments

-Authors provided docking simulations of BLG26 and its potential target protein. At least, in vitro binding assay (e.g., SPR) would be welcomed. If it is not possible, authors should state the limitation in the discussion section.

-Why authors picked up gastric cancer among diverse cancer types? Please describe the rationale briefly in the introduction.

-There are current therapeutic efforts of targeting AKT1? It should be discussed.

-In the molecular docking, how did authors determine the pocket for simulation? 

-The cell lines were verified by using STR?

-The legends inside Fig 3A, B, and C are too small to read.

Author Response

Dear reviewer #3:

Comment 1: Authors provided docking simulations of BLG26 and its potential target protein. At least, in vitro binding assay (e.g., SPR) would be welcomed. If it is not possible, authors should state the limitation in the discussion section.

Response 1: Thanks to you for the constructive comments and suggestions. The kinase assays were than performed to evaluate the inhibition abilities of BLG26 against AKT1. It’s disappointing that BLG26 showed little inhibitory activity to AKT1 kinase. The result was shown below and Ipatasertib was used as positive control. We assume that our compound may inhibited PI3K/AKT/mTOR signal pathway through other mechanisms. In later study, we will focus more attention on it and do further research based on this study.

Fig. S1. Concentration dependent inhibition effects of BLG26 and Ipatasertib on AKT1 kinase. (A) The inhibition effects of compound 6 at the concentration of 1000.0, 500.0, 250.0, 125.0, 62.5, 31.2, 15.6, 7.8, 3.9 nM on kinase AKT1. (B) The inhibition effects of positive control Ipatasertib at the concentration of 1000.0, 500.0, 250.0, 125.0, 62.5, 31.2, 15.6, 7.8, 3.9 nM on kinase AKT1.

Comment 2: Why authors picked up gastric cancer among diverse cancer types? Please describe the rationale briefly in the introduction.

Response 2: Special thanks to you for your valuable suggestion. We tested cytotoxic activities of BLG26 on gastric cancer cell AGS, hepatoma carcinoma cell SMMC-7721, pancreatic cancer cell PANC-1, colorectal cancer cell HCT116. And the results showed that BLG26 had better anti-proliferation activities on gastric cancer cell AGS. We have added explanation for why we picked up gastric cancer for further research in Introduction section.

Comment 3: There are current therapeutic efforts of targeting AKT1? It should be discussed.

Response 3: Thanks to you for constructive comments and suggestions. We have discussed the current therapeutic efforts of targeting AKT1 in Discussion section.

Comment 4: In the molecular docking, how did authors determine the pocket for simulation?

Response 4: Thanks for your kind comments. Binding pockets can be generated automatically based on existing ligands and their positions.

Comment 5: The cell lines were verified by using STR?

Response 5: Thanks for your kind suggestion. The cell lines were bought from ATCC (American type culture collection, AGS Lot, 70012225, PANC-1 70018880, HCT116 70019042) and CCTCC (China center for type culture collection).

Comment 6: The legends inside Fig 3A, B, and C are too small to read.

Response 6: We sincerely appreciate the reviewer’s comments. We have enlarged Fig 3A, B, and C to make sure that it can be clearly read.

Round 2

Reviewer 1 Report

The paper has improved significantly and it is now suitable for publication